# PSME3 regulates migration and differentiation of myoblasts

Kenneth D Kuhn[1,2,3] , Ukrae H Cho[4] , Martin W Hetzer[1]

**The acquisition of cellular identity requires large-scale alterations in cellular state. The noncanonical proteasome activator PSME3 is known to regulate diverse cellular processes, but its importance for differentiation remains unclear. Here, we demonstrate that PSME3 binds dynamically to highly active promoters over the course of differentiation. However, loss of PSME3 does not globally affect mRNA transcription. We find instead that PSME3 influences the levels of several adhesion-related proteins and acts upstream of the HSP90 co-chaperone NUDC to regulate cell motility and myoblast differentiation in a proteasome-independent manner. Our findings reveal several new facets of PSME3 functionality and highlight its importance for the differentiation of myogenic cells.**

## Introduction

Differentiation requires the coordinated restructuring of diverse cellular systems. Critical to this process are changes in nuclear organization, which have long been known to be influenced by the activity of the proteasome (McCann & Tansey, 2014). Indeed, the proteasome has such an intimate role in chromatin dynamics, including the modification of histones and initiation of transcription, that one of its components was originally mistaken for a member of the RNA polymerase II (RNAPII) holoenzyme (Kim et al, 1994; Muratani & Tansey, 2003). The proteasome consists of a proteolytic core capped on either end by a regulatory subunit, the best-studied of which is the ubiquitin-binding 19S complex. However, other subunits have been found to play similarly important roles in regulating nuclear processes. Among these is proteasome activator complex subunit 3 (referred to here as PSME3, though also known as PA28γ, 11S REGγ, or Ki nuclear autoantigen) (Cascio, 2021).

PSME3 is a nuclear, homoheptameric protein whose function has been linked to the organization of the nuclear speckle, Cajal body, and promyelocytic leukemia body (Cioce et al, 2006; Baldin et al, 2008; Zannini et al, 2009; Jonik-Nowak et al, 2018). PSME3 has also been shown to interact with several important regulators of the cell cycle including p16, p19, and p21, as well as the transcriptional regulators c-Myc, KLF2, SMURF2, NF-κB, and LATS1/2 (Chen et al, 2007; Nie et al, 2010; Li et al, 2015; Sun et al, 2016; Xu et al, 2016; Wang et al, 2018). Because of these functions, loss of PSME3 is associated with G1 arrest in cultured cells and impaired growth rates in mice (Murata et al, 1999; Masson et al, 2001; Barton et al, 2004; Chen et al, 2017). Conversely, PSME3 is frequently overexpressed in various cancer cell lines and associated with accelerated cell division, metastatic potential, and reduced rates of apoptosis (Mao et al, 2008; Lei et al, 2020). Though PSME3 is often understood to operate through the degradation of target proteins, only a small portion of the PSME3 population is associated with the core proteasome (Fabre et al, 2014; Welk et al, 2016). In fact, several of its functions are carried out in a proteasome-independent manner. Through an unknown mechanism, PSME3 can induce mitotic arrest, regulate p53 levels, and maintain global chromatin compaction even when prevented from interacting with the proteasome (Zannini et al, 2008; Zhang & Zhang, 2008; Fesquet et al, 2021).

Although PSME3 performs diverse functions in systems critical for differentiation, the mechanisms by which it drives cell identity acquisition are only beginning to be understood. It was recently found that loss of PSME3 impairs T-cell maturation and triggers the differentiation of Th17 cells by altering the cell surface protein profile of dendritic cells (Barton et al, 2004; Zhou et al, 2020). In addition, suppression of PSME3 expression biases bone marrow stromal cells toward an adipogenic rather than osteogenic fate, and mice lacking PSME3 display corresponding bone-healing defects (Chen et al, 2024). Given these findings, we hypothesized that PSME3 may be important for the differentiation of other cell types, such as those found in the muscular system.

To investigate this possibility, we used C2C12 myoblasts, which can be differentiated into syncytial myotubes upon withdrawal of serum, to interrogate multiple stages of differentiation (Yaffe & Saxel, 1977; Blau et al, 1983). We find for the first time that PSME3 binds extensively to the chromatin at highly active promoter regions, likely through an association with the protein RNAPII

[1]Institute of Science and Technology (ISTA), Klosterneuburg, Austria   [2]Molecular and Cell Biology Laboratory, Salk Institute for Biological Studies, La Jolla, CA, USA   [3]The Neurosciences Graduate Program, University of California San Diego, La Jolla, CA, USA   [4]Department of Cell Biology and Physiology and Lineberger Comprehensive Cancer Center, UNC Chapel Hill, Chapel Hill, NC, USA

Correspondence: martin.hetzer@ist.ac.at

regulator RPRD1A. Surprisingly, however, PSME3 depletion has no global effect on gene expression. Instead, we discover that PSME3 interacts with HSP90 co-chaperone NUDC and regulates the levels of cell adhesion– and migration-related proteins. As a result, myoblasts lacking PSME3 display accelerated migration and impaired myogenesis. Finally, we show that although PSME3 requires NUDC to perform these functions, it acts independently of the proteasome, thereby establishing PSME3 as a novel regulator of myoblast differentiation.

# Results

## PSME3 binds to highly active promoters before differentiation

Previous studies have established a role of PSME3 in maintaining the integrity of a variety of nuclear structures. In particular, PSME3 was found to associate with several heterochromatic regions and maintain their compacted state (Fesquet et al, 2021). We sought to extend this work by globally investigating the binding of PSME3 to genomic regions across differentiation. To test whether PSME3 associates with chromatin in C2C12 myoblasts, we performed Cleavage Under Targets & Release Using Nuclease (CUT&RUN) against the endogenous population of PSME3. Surprisingly, we found that PSME3 associates extensively with the chromatin at over 5,000 distinct regions in cycling myoblasts (Fig 1A and B). A large majority of these peaks were found to be co-positive for the active promoter mark H3K4me3. Furthermore, PSME3 showed an even stronger preference for promoter-proximal regions (<= 1 kb) than H3K4me3, mainly at the expense of binding to gene bodies or intergenic regions (Fig 1C). Of all transcriptionally active promoters, those bound by PSME3 are on average roughly 50% more active than those that lack PSME3 (Fig 1D).

To determine whether PSME3 exchanges its binding sites as differentiation progresses, we performed CUT&RUN in myoblasts that had differentiated for 2 d. In contrast to cycling cells, Day 2 cells displayed a near-complete absence of PSME3 at the chromatin (Fig 1E). Even when subjected to mild formaldehyde fixation to reduce the lability of transient chromatin interactions, scarcely any increase in the number of PSME3 peaks was observed in cells assayed, analyzed either fresh or after freezing (Fig S1). In summary, PSME3 binds selectively to highly active promoters in myoblasts, but becomes undetectable by the second day of myogenic differentiation.

## PSME3 interacts with RPRD1A but does not regulate gene expression

To understand what function PSME3 might be serving at the chromatin, we performed co-immunoprecipitation on endogenous PSME3 to identify potential interaction partners (Fig 2A). Among the most enriched binding partners were PSME3-interacting protein PSME3IP1, a known PSME3 interactor (Jonik-Nowak et al, 2018), and RPRD1A (Fig 2A and B). RPRD1A regulates the dephosphorylation of S5 of RNAPII's C-terminal domain to facilitate progression of the polymerase from the promoter into the gene body (Ni et al, 2014; Ali

et al, 2019). Given that S5P is found on polymerases poised to begin transcription and can recruit methyltransferases to deposit H3K4me3 marks on active promoters (Ng et al, 2003), we hypothesized RPRD1A was a potential mediator for PSME3's chromatin-binding activity.

To verify this interaction in situ, we used a proximity ligation assay (PLA), which produces a fluorescent signal at points where two proteins are removed by a distance of <40 nm. Cycling C2C12 cultures were transfected with a plasmid expressing FLAG-PSME3, and the colocalization of FLAG with RPRD1A was assessed (Fig 2C). Cells expressing FLAG-PSME3 showed a high level of PLA signal, whereas nontransfected cells showed little to no signal. Omission of either antibody similarly abolished the signal (Fig S2A and B), indicating its specificity.

As alterations in both RPRD1A and PSME3 function have previously been demonstrated to affect gene expression, we asked whether PSME3 may be a regulator of transcription during differentiation (Li et al, 2006; Wu et al, 2010; Sun et al, 2016; Xu et al, 2016; Wang et al, 2018; Bhatti et al, 2019). We performed RNA sequencing on PSME3-deficient cells at three stages of differentiation: before differentiation (cycling), fully confluent cells immediately before the withdrawal of serum (Day 0), and cells that have differentiated for 2 d and begun to form myotubes (Day 2). At each of these time points, we observed no global change in gene expression in cells depleted of PSME3 (Fig 2D). Taken together, these results suggest that PSME3 interacts with RPRD1A in C2C12 myoblasts, but does not regulate transcription during myogenesis.

## PSME3 regulates cell migration rates and levels of adhesion proteins

During the course of our experiments, we noticed that cells lacking PSME3 were more difficult to detach from the culture plates with trypsin, indicating a greater level of adhesion (data not shown). We therefore turned our attention to the next most enriched protein in the PSME3 immunoprecipitate, NUDC (Fig 2B). NUDC was recently identified as a co-chaperone of HSP90 and has been documented to regulate cell migration through the stabilization of several cytoskeletal proteins related to cell adhesion and migration, including cofilin and filamin A (Zhang et al, 2016; Islam et al, 2020; Liu et al, 2021; Biebl et al, 2022).

The observed interaction between NUDC and PSME3 was unexpected, as NUDC is considered to be a predominantly cytoplasmic protein (Zhou et al, 2003; Zhu et al, 2010; Zhang et al, 2016; Islam et al, 2020), whereas PSME3 is nuclear but may relocalize to the cytoplasm under certain conditions (Zannini et al, 2008; Kobayashi et al, 2013; Pecori et al, 2021; Carrettiero et al, 2022). To better understand the spatial overlap between the two proteins in our system, we first stained cycling C2C12 cells for endogenous PSME3 and found it to be localized to the nucleus (Fig S3A). We in addition performed live imaging on cycling C2C12 cells that were transfected with constructs expressing PSME3 tagged with either N- or C-terminal GFP, confirming PSME3's presence in the nucleus (Fig S3B). Furthermore, we found that NUDC displayed a strong signal across both cytoplasm and nucleus, where it overlapped with PSME3 staining (Fig S3C), indicating that they are well positioned to interact. To verify this interaction, we performed PLA using antibodies

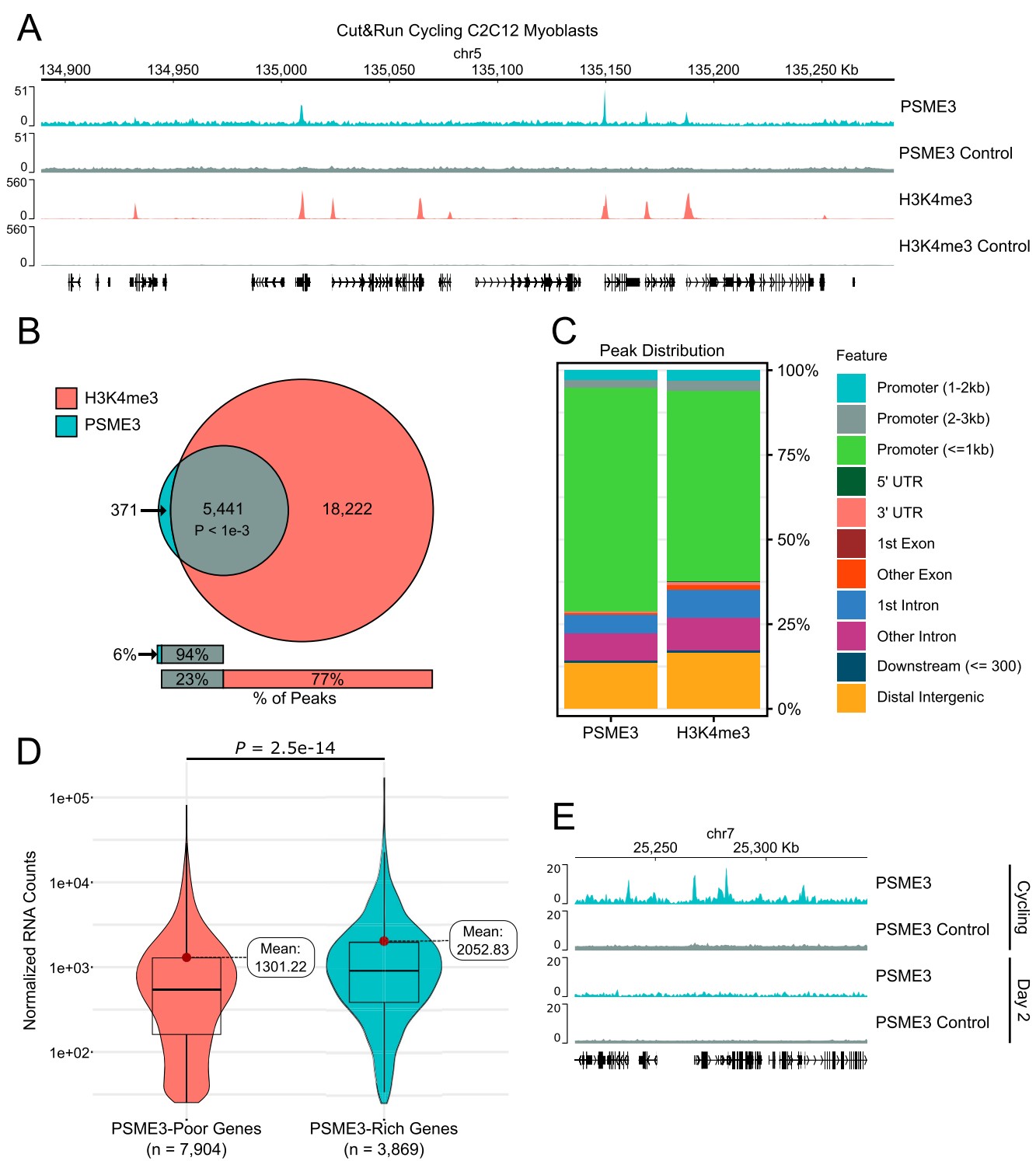

**Figure 1. PSME3 dynamically binds to highly active promoters in cycling myoblasts.**
**(A)** CUT&RUN was performed in cycling C2C12 cells using antibodies targeting PSME3 or H3K4me3 to measure their position on the chromatin. Three biological replicates were performed and are represented here in aggregate. **(B)** Peaks were called using MACS2 for both PSME3 and H3K4me3, and their colocalization was measured. The likelihood of this level of colocalization by chance alone was determined by a permutation test with 1,000 permutations (Gel et al, 2016); P-value achieved the minimum possible value of $9.99 \times 10^{-4}$ and z-score of 403.961. **(C)** PSME3 and H3K4me3 CUT&RUN peaks were annotated with ChIPseeker for the type of region in which they reside. **(D)** RNA sequencing was performed on cycling C2C12 cells. Active genes that possess a PSME3 peak were separated from those that did not, and expression levels were analyzed. Data are collected with three biological replicates per condition and were analyzed with a Welch two-sample $t$ test. **(E)** CUT&RUN was performed additionally in Day 2 differentiated C2C12 cells using an antibody targeting PSME3. **(A)** Each track is the result of a single replicate, and the cycling replicate is drawn from those represented in panel (A). Source data are available for this figure.

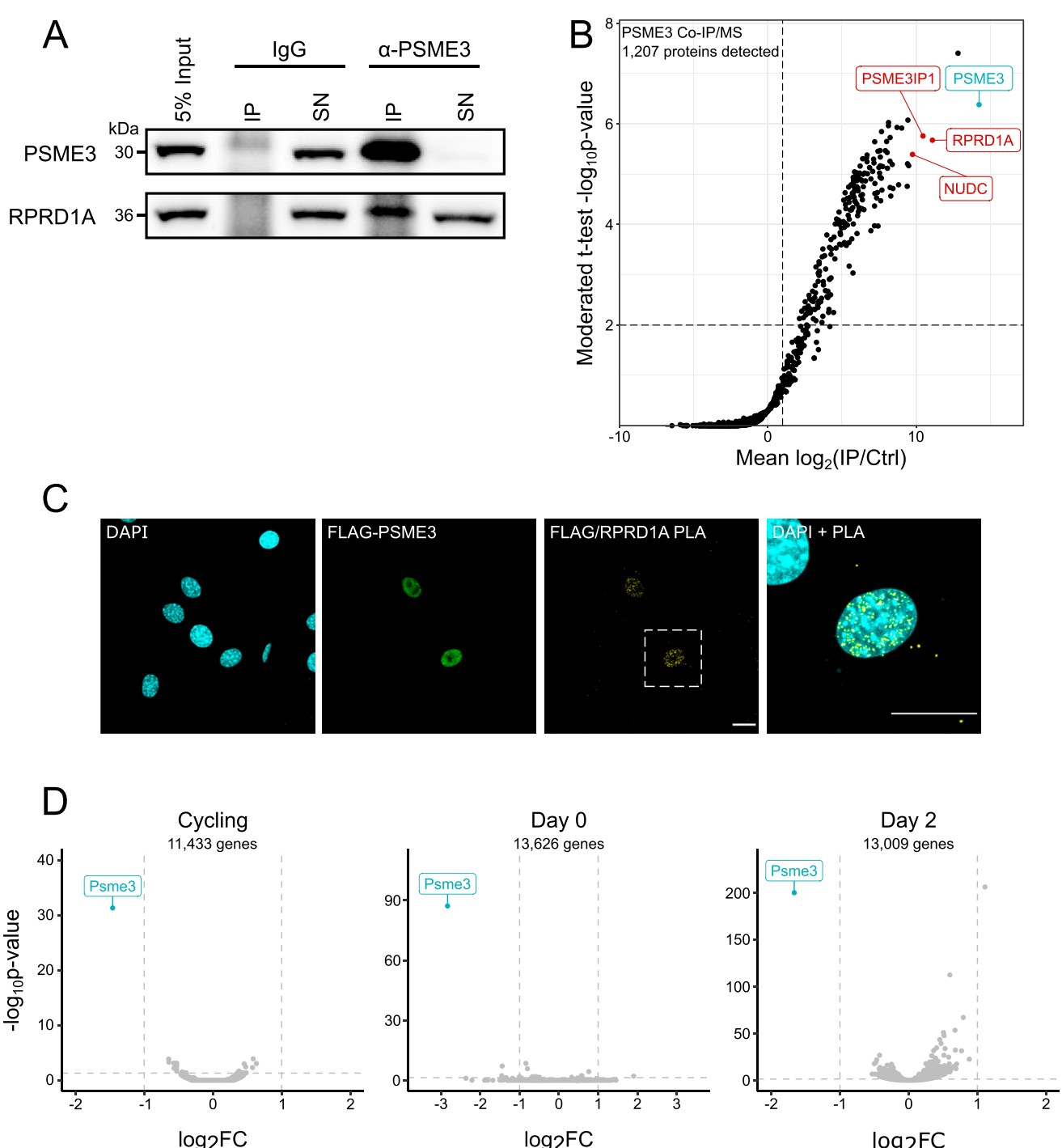

**Figure 2. PSME3 interacts with RPRD1A without affecting global gene expression.**
**(A)** PSME3 was precipitated from C2C12 cells while cycling and at Days 0 and 1 of differentiation using a PSME3-specific antibody. Efficiency of PSME3 and RPRD1A capture from Day 0 lysates compared with the supernatant (SN) versus that performed with an IgG control was assessed by a Western blot using antibodies targeting PSME3 or RPRD1A. **(B)** Label-free quantitative mass spectrometry was performed on the PSME3 coprecipitate across all three time points. **(C)** Cycling C2C12 cells expressing FLAG-PSME3 were subjected to proximity ligation assay using primary antibodies targeting either FLAG or RPRD1A. The scale bar is 20 microns in length. **(D)** RNA collected from Cycling, Day 0, or Day 2 C2C12 cells lacking PSME3 was sequenced and compared to cells treated with scrambled siRNA.
Source data are available for this figure.

targeting FLAG-PSME3 and NUDC, which specifically revealed the association of the two proteins within the nucleus (Fig 3A and S4A and B).

To test for a functional interaction between PSME3 and NUDC, we measured the migration rates of fully confluent Day 0 myoblasts in a scratch assay. Cells lacking PSME3 showed a significant increase

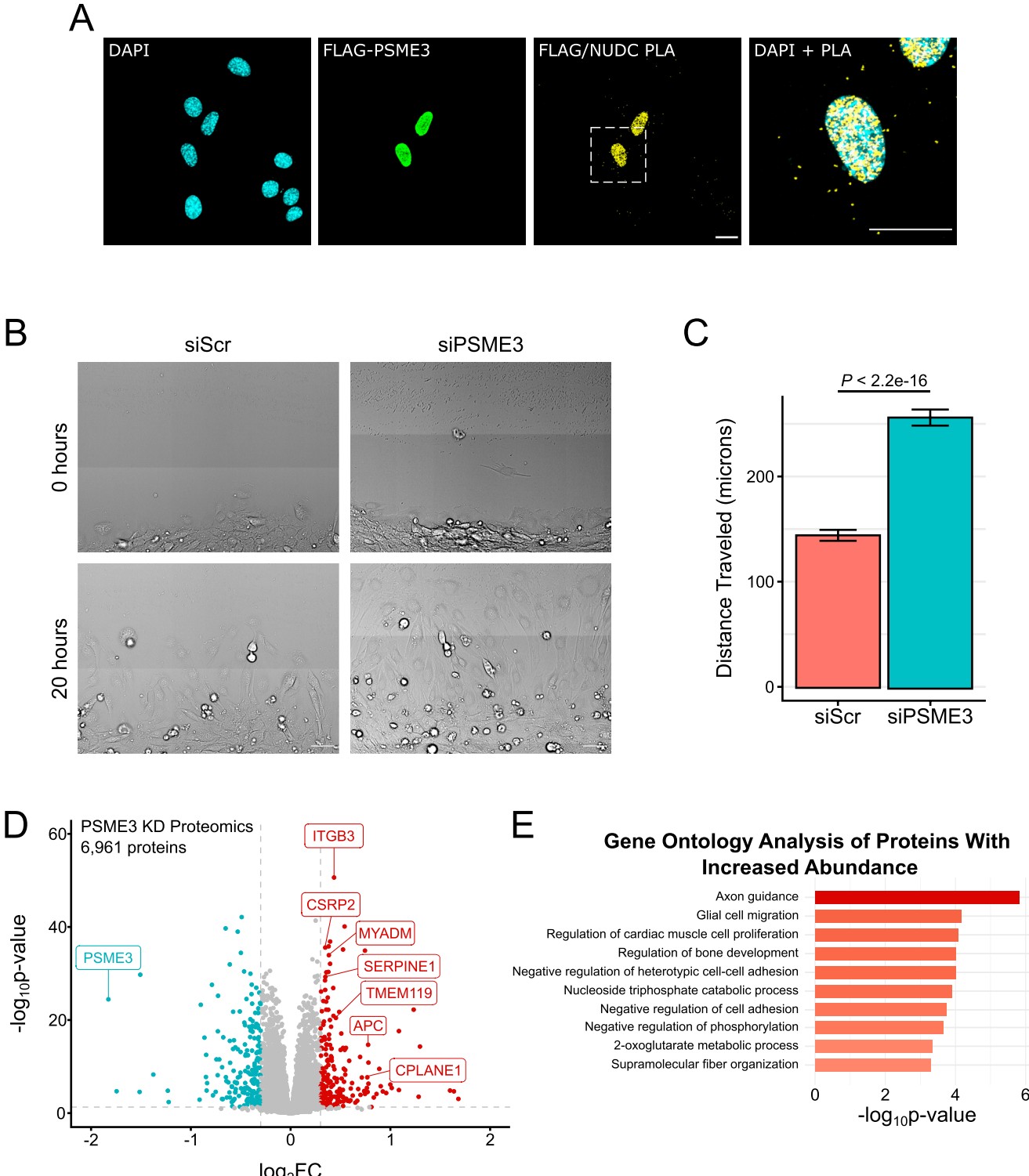

**Figure 3. PSME3 associates with NUDC and regulates cell migration.**
**(A)** PLA was performed in cycling C2C12 cells expressing FLAG-PSME3 using antibodies against FLAG and NUDC, and with the respective omission of either or both antibodies as controls. The scale bar is 20 microns in length. **(B)** Day 0 confluent C2C12 cells were scratched and allowed to migrate for 20 h while undergoing continuous brightfield imaging; the scale bar is 50 microns in length. **(C)** Quantification of the distance traveled over 20 h by individual migrating cells treated with either scrambled (Scr) siRNA or that targeting *Psme3*. Data are collected with three biological replicates, each composed of two technical replicates of *n* = 15 each, and were analyzed by a Welch two-sample *t* test; error bars show the standard error of the mean. **(D)** Day 0 confluent C2C12 cells treated with either scrambled or PSME3-targeting siRNA were subjected to label-free proteomics. The changes in protein abundance are plotted here with decreased-abundance proteins in blue and increased-abundance proteins in

in migration rate over the control group, consistent with an alteration in cytoskeletal behavior (Fig 3B and C). To determine the molecular basis for this change, we performed proteomics on whole-cell lysates from these same cultures. We discovered increased abundance of several cell adhesion proteins upon *Psme3* knockdown, including the Rac1 regulator myeloid-associated differentiation marker (MYADM) and integrin beta-3 (ITGB3) (Fig 3D). Gene ontology (GO) analysis of differentially regulated proteins revealed an increase in species related to cytoskeletal organization, as well as cell migration and adhesion (Fig 3E). Taken together, these results indicate that PSME3 interacts with NUDC and regulates both rates of cell migration and levels of cell adhesion proteins.

### Loss of PSME3 impairs myotube formation

We reasoned that if PSME3 affects cell migration, it may also be critical for the differentiation of C2C12 myoblasts, which are particularly sensitive to changes in the cytoskeletal system because of the heavy demands imposed by membrane apposition and cell fusion (Lehka & Rędowicz, 2020). To this end, we depleted C2C12 myoblast cells of PSME3 through siRNA treatment before serum withdrawal and subsequent differentiation (Fig 4A and B). siRNA-treated myoblasts were allowed to differentiate for two days before fixation and staining with antibodies against myogenin, a myogenic transcription factor, and myosin heavy chain (MHC), a major contractile protein found in differentiated myotubes. Widefield imaging revealed a clear deficit in differentiation (Fig 4C); cultures lacking PSME3 produced myotubes with fewer nuclei (Fig 4D) and a lower fusion index (Fig 4E), defined as the percentage of nuclei contained within MHC-positive myotubes.

To determine whether the effect of PSME3 on myogenesis is dependent on an interaction with the proteasome, we stably expressed FLAG-tagged PSME3 either in its full length or with a deletion of the terminal 14 amino acids, which renders it unable to associate with the 20S proteasome (Fig S5B) (Ma et al, 1993; Förster et al, 2005; Zannini et al, 2008; Zhang & Zhang, 2008; Fesquet et al, 2021). We found that the expression of either the full-length or C-terminal deletion mutant was sufficient to rescue the myogenesis phenotype caused by depletion of endogenous PSME3, indicating that PSME3 regulates myoblast differentiation through a proteasome-independent mechanism (Fig S5A–C).

Though there were no global alterations in gene expression when PSME3 was depleted, we observed a modest increase in several collagen transcripts on the second day of differentiation (Fig 2D). To confirm that the observed differentiation phenotype is the result of a cell-intrinsic effect rather than an alteration in the extracellular matrix, we created a mixed culture system of cells treated with scrambled siRNA or that targeting *Psme3* and labeled them with a red or blue dye, respectively. If the differentiation phenotype is due to a cell-extrinsic effect, the presence of control cells should improve the differentiation efficiency of those lacking PSME3. However, when the mixed population was induced to differentiate for three days, we observed a selective exclusion of cells lacking PSME3 from mature myotubes (Fig S5D and E). These results indicate that PSME3 is necessary for myoblast differentiation in a cell-intrinsic manner.

### PSME3 regulates differentiation and migration upstream of NUDC

To better understand how PSME3 and NUDC coordinate to regulate differentiation in C2C12 cells, we generated cells deficient in either PSME3, NUDC, or both (Fig 5A). We suspected that PSME3 may negatively regulate NUDC's ability to stabilize cell adhesion– and migration-related proteins, which would be consistent with our observation that loss of PSME3 increases migration rates (Fig 3B and C). If PSME3 acts upstream of NUDC, loss of PSME3 in a NUDC-deficient background should have no effect on the rates of cell migration or differentiation.

In support of this hypothesis, we found that although NUDC-deficient cells showed decreased migration rates, additional loss of PSME3 showed no positive effect on cell motility (Fig 5B and C). Furthermore, loss of NUDC produced a differentiation defect that is unaffected by additional depletion of PSME3, indicating that PSME3 is likely unable to exercise any pro-differentiation activity in the absence of NUDC (Fig 5D and E). Taken together, these data support the hypothesis that PSME3 acts upstream of NUDC in the regulation of migration and differentiation.

## Discussion

PSME3 is an elusive and versatile protein possessed of myriad functions, many of which remain to be uncovered. In this study, we reveal several new facets of PSME3's activity. As the first unbiased study into PSME3's genome binding properties, we demonstrate a previously unrecognized capacity of PSME3 to associate with the RNAPII regulator RPRD1A and highly active promoters. Furthermore, we show a novel interaction of PSME3 with the HSP90 co-chaperone NUDC, through which it may regulate cell migration rates, the levels of cell adhesion–related proteins, and myogenesis.

Though the 19S-containing proteasome has long been known to associate with the chromatin, particularly at highly active gene regions (Auld et al, 2006), the finding that the alternative proteasome cap PSME3 possesses a similar capacity is a novel discovery. We believe that this association may be mediated by an interaction with RPRD1A, which regulates the posttranslational modification state of RNAPII and localizes to the promoters of active genes (Liu et al, 2015). This aligns well with our finding that PSME3 mirrors H3K4me3 peaks in topology and localization and is enriched at the most transcriptionally active promoters. This discovery, in combination with a recent study by Fesquet and colleagues showing that PSME3 binds to and is crucial for maintaining heterochromatic regions (Fesquet et al, 2021), suggests that PSME3 possesses broad chromatin-binding activity.

red, with several select species involved in cell migration receiving a label. Five biological replicates are represented here together. **(E)** Gene ontology analysis of increased-abundance proteins; the top 10 categories are shown.
Source data are available for this figure.

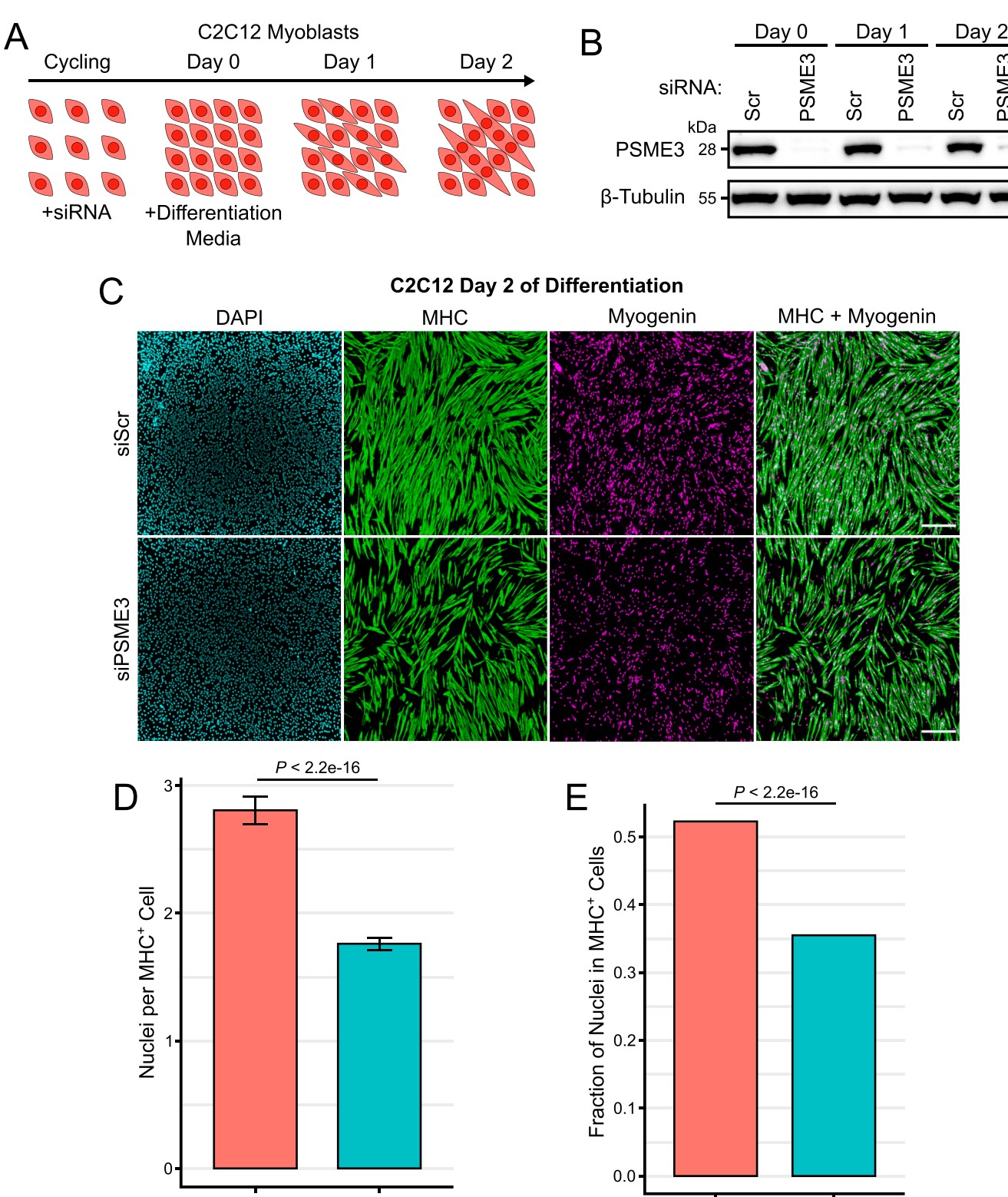

**Figure 4. Depletion of PSME3 impairs myotube formation.**
**(A)** Cycling C2C12 cells were treated with siRNA before growth to confluency and serum deprivation, which together induce differentiation and myotube formation.
**(B)** C2C12 cells were treated with scrambled (Scr) or *Psme3*-targeting siRNA and collected at Days 0, 1, 2 of differentiation; target protein levels were assessed by Western blot. **(C)** C2C12 cells depleted of PSME3 were induced to differentiate for two days and were subjected to immunofluorescence using antibodies targeting MHC and myogenin; the scale bar is 200 microns in length. **(D)** Immunostained cultures were analyzed for the number of nuclei contained in each MHC-positive cell. Data include

We were surprised, however, to see very little change in gene expression upon PSME3 depletion. PSME3 has previously been demonstrated to influence the expression of individual genes in various cell types (Li et al, 2006; Sun et al, 2016; Xu et al, 2016; Wang et al, 2018; Bhatti et al, 2019), though its loss has no effect on bulk RNA production (Cioce et al, 2006; Baldin et al, 2008). Furthermore, many of the transcriptional regulators documented to be influenced by PSME3, such as p53, MAPK, and YAP1, were unaffected in abundance by the loss of PSME3 in C2C12 myoblasts (Fig 3D). Why PSME3 should occupy highly active promoter regions despite having no apparent role in their regulation is unclear at this point. It is possible that PSME3 may be primed to perform some function that is not engaged during the course of differentiation. For example, it is well established that double-strand DNA breaks (DSBs) induce global degradation of RNAPII complexes in a proteasome-dependent manner (Steurer et al, 2022). Furthermore, loss of PSME3 has been found to sensitize cells to radiomimetic treatment and prevent timely repair at DSB sites (Levy-Barda et al, 2011). Thus, it is possible that PSME3 plays a role in degrading RNAPII in the case of DNA damage. Alternatively, PSME3 may respond to other types of cellular insults, such as it has been shown to do in the context of hyperosmotic stress (Carrettiero et al, 2022), and under other such circumstances is perhaps capable of adopting a gene regulatory role.

In addition to DNA binding, we observed an interaction between PSME3 and the co-chaperone NUDC, which mediates cargo transfer between HSP70 and HSP90 (Biebl et al, 2022). NUDC was originally identified in the filamentous fungus *Aspergillus nidulans* for its role in regulating the microtubule-based migration of nuclei within the cell (Osmani et al, 1990; Fu et al, 2016). In addition, a role of NUDC in regulating cell migration has been demonstrated in diverse cell types through stabilization of several migration-related proteins (Aumais et al, 2001; Cappello et al, 2011; Zhang et al, 2016; Islam et al, 2020; Liu et al, 2021).

We observed that depletion of PSME3 in C2C12 cells resulted in an increase in the rate of cell migration and altered the abundance of cell adhesion–related proteins, including ITGB3 and the Rac1 regulator MYADM. ITGB3 has been shown to regulate cell adhesion and migration in response to diverse ligands, and its expression in muscle satellite cells influences both migration rates and differentiation efficiency (Byzova et al, 2000; Bax et al, 2003; Lei et al, 2011; Liu et al, 2011; Zhu et al, 2019). In addition, ITGB3 and MYADM are regulators of Rac1, which has been shown to be crucial for migration, adhesion, and membrane apposition, such that loss of Rac1 effectively abolishes myotube formation (Vasyutina et al, 2009; Aranda et al, 2011; Liu et al, 2011). Because the proper balance of migration regulators is critical for myogenesis, we propose that the observed increased expression of ITGB3, MYADM, and other such regulators upon PSME3 depletion results in the reduced differentiation efficiency observed in our system (Lehka & Rędowicz, 2020).

In support of this hypothesis, we find that PSME3's influence on cell migration is dependent on the presence of NUDC, such that loss of PSME3 has no positive effect on cell migration in the absence of NUDC. This is consistent with a model in which PSME3 antagonizes NUDC-mediated protein stabilization, maintaining a physiological level of migration-related proteins. Although we establish PSME3's independence of the proteasome for differentiation, how PSME3 might interact with NUDC to achieve these functions remains unclear. PSME3 has been observed to facilitate protein degradation through the recruitment of ubiquitination factors even when it is incapable of directly binding the proteasome (Zhang & Zhang, 2008). One possibility is that PSME3 similarly facilitates the degradation of target proteins of NUDC, which may itself aid in the handoff of protein cargo from the chaperone system to the proteasome. This role has been demonstrated for several members of the Bcl2-associated athanogene (BAG) family of proteins, including BAG2, which is known to associate with both HSP70 and PSME3 under stress conditions (Abildgaard et al, 2020; Carrettiero et al, 2022).

In summary, our results establish PSME3 as an important regulator of myogenesis and will facilitate future research into the contribution of this protein to the acquisition of cell identity.

# Materials and Methods

## Plasmids

Constructs were created by insertion of *Psme3* into pcDNA3.1(+)-C-eGFP or pcDNA3.1(+)-N-eGFP for live imaging, and pcDNA3.1(+)-N-DYK or pcDNA3.1(+)-C-DYK for FLAG-tagged experiments. The *Psme3* sequence (accession NM_011192.4) was obtained from GenScript.

## Antibodies

The following antibodies were used: PSME3 (rabbit polyclonal, BML-PW8190, 1:6,000 IF, 1:200 CUT&RUN; Enzo Life Sciences) (rabbit polyclonal, 38-3800, 1:125 WB, 1:25 IP; Thermo Fisher Scientific); β-tubulin (rabbit monoclonal 9F3, #2128, 1:1,000 Western blot; Cell Signaling Technology); myosin-4 (mouse monoclonal MF20, 14-6503-82, 1:100 IF; Thermo Fisher Scientific); myogenin (mouse monoclonal F5D, 14-5643-82, 1:250 IF; Thermo Fisher Scientific); FLAG (mouse monoclonal M2, F1804, 1:500 IF and PLA, 1:1,000 Western blot; Sigma-Aldrich); RPRD1A (rabbit polyclonal, HPA040602, 1:50 PLA, 1:250 Western blot; Sigma-Aldrich); NUDC (rabbit polyclonal, 10681-1-AP, 1:800 Western blot, 1:25 IF and PLA; Proteintech); H3K4me3 (rabbit monoclonal C42D8, #9751, 1:50 CUT&RUN; Cell Signaling Technology); H3 (mouse monoclonal 1B1B2, #14269, 1:1,000 Western blot; Cell Signaling Technology).

three biological replicates, each comprising two technical replicates of $n$ = 250 cells, and were analyzed by a Welch two-sample $t$ test with error bars showing the standard error of the mean. **(E)** Immunostained cultures were analyzed for the percentage of all DAPI-positive nuclei that are contained within MHC-positive cells. Data include three biological replicates each comprising two technical replicates with an $n$ of roughly 20,000 cells each, and are analyzed by a two-sample $t$ test of proportions. Source data are available for this figure.

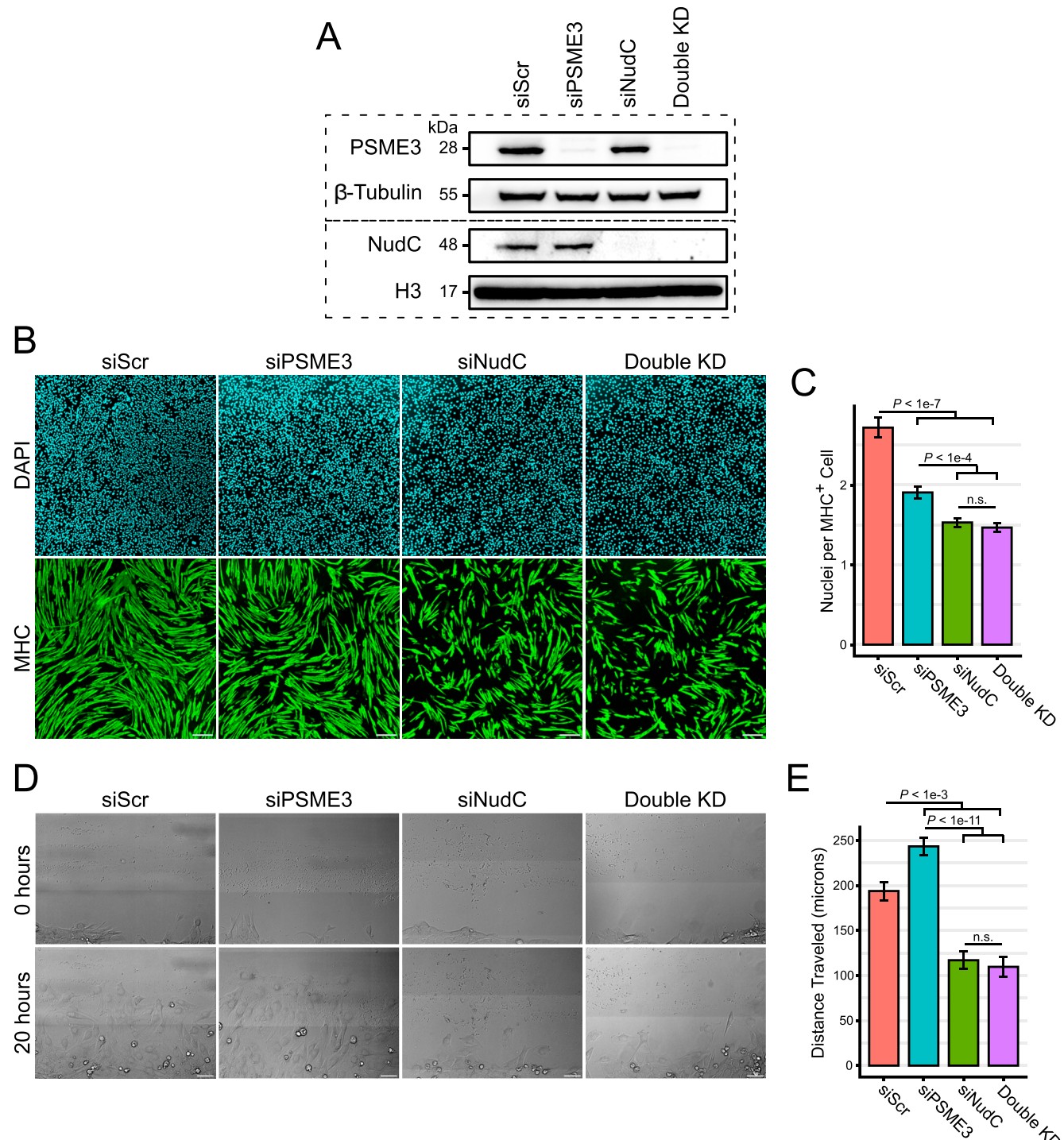

**Figure 5. PSME3 acts upstream of NUDC in the regulation of migration and differentiation.**
**(A)** C2C12 cells were treated with scrambled siRNA (Scr), *Psme3*-targeting siRNA, *NudC*-targeting siRNA, or a combination of both *Psme3*- and *NudC*-targeting siRNA to produce a double knockdown (Double KD). All samples received an identical amount of total siRNA; see methods for details. Cell lysates were collected at Day 0, and target protein levels were assessed by Western blot. Dotted lines indicate separate blots performed with the same samples accompanied by independent loading controls. **(B)** siRNA-treated cells were induced to differentiate for two days and were subjected to immunofluorescence using antibodies targeting MHC; the scale bar is 200 microns in length. **(C)** Immunostained cultures were analyzed for the number of nuclei contained in each MHC-positive cell. Data include two technical replicates, each composed of *n* = 250 cells, and were analyzed by a Welch two-sample *t* test with error bars showing the standard error of the mean. **(D)** Day 0 confluent siRNA-treated C2C12 cells were scratched and allowed to migrate for 20 h while undergoing continuous brightfield imaging; the scale bar is 50 microns in length. **(E)** Quantification of the distance traveled over 20 h by individual migrating cells under each treatment condition. Data are derived from two technical replicates of *n* = 15 each and were analyzed by a Welch two-sample *t* test; error bars show the standard error of the mean.
Source data are available for this figure.

## Cell culture and transfection

C2C12 cells obtained from the American Type Culture Collection (ATCC) were cultured in DMEM (high glucose, pyruvate, 11995065; Thermo Fisher Scientific) with 20% FBS. Cells were allowed two passages (4 d) after thawing before being subjected to any experiments. No cells over passage nine were used for any experiment, and the cells were not allowed to exceed 70% confluency at any time before differentiation.

For transfection with siRNA, cycling C2C12 cells were reverse-transfected on two consecutive days with Lipofectamine RNAiMAX transfection reagent (Thermo Fisher Scientific) and either SMARTpool siRNA from Horizon Discovery targeting *Psme3* (L-062727-01-0005), *NudC* (L-041603-01-0005), or a nontargeting control pool (DH-D-001810-10-20) in Opti-MEM at 50 nM as described below.

On the first day of transfection, 15 µl of 20 µM siRNA solution and 15 µl of Lipofectamine were combined in 1.5 ml of Opti-MEM in a 6-cm dish (31985062; Thermo Fisher Scientific) and incubated for 20 min at room temperature. C2C12 cells were trypsinized from a separate plate, and 300,000 cells suspended in 4.5 ml of growth media were added to the Opti-MEM solution. On the next day, 30 µl of 20 µM siRNA solution and 30 µl of Lipofectamine were combined in 2.4 ml of Opti-MEM in a 10-cm dish and incubated for 20 min at room temperature. 300,000 siRNA-treated C2C12 cells from the previous day were resuspended in 9 ml of growth media and added to the Opti-MEM solution. Cells were allowed to recover for 2 d before being collected or induced to differentiate. For differentiation into myotubes, cells were grown to full confluency, washed once with PBS, and switched to DMEM with 2% horse serum, designated as Day 0. Differentiation media were refreshed every 48 h.

For double-knockdown experiments shown in Fig 5, cells were treated exactly as above, but with a change in reagent concentration. Each condition received siRNA as follows: siScr received 75 nM scrambled siRNA; siPSME3 received 37.5 nM *Psme3*-targeting siRNA and 37.5 nM scrambled siRNA; siNUDC received 37.5 nM *NudC*-targeting siRNA and 37.5 nM scrambled siRNA; and double knockdown (DKD) received 37.5 nM *Psme3*-targeting siRNA and 37.5 nM *NudC*-targeting siRNA. The volume of Lipofectamine RNAiMAX transfection reagent was similarly increased to 22 and 44 µl on the first and second day of transfection, respectively, as was the volume of Opti-MEM similarly increased to 2 and 3 ml on the first and second day of transfection, respectively, to accommodate the increased amount of siRNA.

For transfection with plasmids, cells were grown to 80% confluency and were treated with a mixture of Lipofectamine 2000 (11668019; Thermo Fisher Scientific) and the plasmid of interest. Cells were passaged on the next day and examined on the day thereafter for expression.

## Immunofluorescence and PLA

Two methods were used for immunofluorescence. In the first method, cells were fixed with 4% formaldehyde in PBS for 5 min at room temperature. They were then incubated with a blocking/permeabilization buffer containing 0.1% Triton X-100, 0.02% SDS, and 10 mg/ml BSA in PBS. The cells were incubated in primary antibody in the blocking solution for 90 min, followed by three washes in PBS, an incubation with fluorescent secondary antibodies in the blocking solution, then an additional three washes in PBS before mounting in Everbrite Mounting Medium with DAPI (23002; Biotium). This method was used for all imaging experiments that did not involve staining for NUDC.

In the second method, fixation was performed by incubation with 4% formaldehyde in PBS for 5 min at room temperature, followed by 8 min in cold 100% methanol at −20°C. The blocking/permeabilization buffer was composed of 0.1% saponin with 10 mg/ml BSA, which was also used as a diluent for the primary and secondary antibody. This method is otherwise identical to the first method and was used exclusively for experiments requiring staining of NUDC.

The PLA was performed with DuoLink In Situ Orange Starter Kit (DUO92102-KT; Sigma-Aldrich) according to the manufacturer's instructions with the following exceptions: permeabilization/blocking and primary antibody dilution/incubation were performed with the buffers and according to the principles described below. Negative controls included several wells in which one or both of the primary antibodies were omitted, while all other components remained.

Imaging of mature myotubes and cell migration was performed with a Nikon Ti2-E Widefield Fluorescence Microscope using a 20x objective, whereas all other imaging experiments used a Leica SP8 Laser Confocal Microscope using a 63x oil immersion objective. All imaging experiments were performed in *µ*-Slide 8 Well ibiTreat ibidi chambers.

## Retrovirus production

The coding sequence of *Psme3* was FLAG-tagged and cloned into the pQCXIB vector, with which 80% confluent HEK293T cells were transfected along with the pCL-Ampho retrovirus packaging vector in equal proportions. The media were changed at 24 h after transfection, and virus-containing conditioned media were collected an additional 24 h thereafter. After being passed through a 0.45-micron filter, the conditioned media were combined 1:1 with fresh growth media and incubated with C2C12 cells for 24 h before exchange with fresh growth media. At 48 h after infection, cells were selected with 1–2 µg/ml of puromycin until cell death ceased and control cells had uniformly perished. For knockdown experiments, a pool of custom-designed siRNA from Horizon Discovery targeting the 3'-UTR of *Psme3* was used to deplete the endogenous copy of *Psme3* while sparing the tagged variants.

## Cell mixing

Cells were treated with siRNA as described above, and on the day before differentiation were stained with either CellTracker Deep Red dye (1 mM solution at 1:250, C34565; Thermo Fisher Scientific) or CellTrace Violet dye (5 mM at 1:500, C34571; Thermo Fisher Scientific) according to the manufacturer's instructions and as previously described (Zhang et al, 2017). After staining, cells were counted and plated either in isolation or in a 1:1 mixture. Differentiation was induced on the next day, and incorporation into myotubes was

assessed on Day 3 after fixing and staining for MHC. Because of the level of cell death normally observed during differentiation, blue nuclei indicating cells lacking PSME3 were counted by hand. The fraction contained in MHC-positive structures was quantified and compared between the separate conditions.

### Image analysis

Cell migration was measured manually by displacement of the center of the nucleus over 2-h intervals. The sum of these displacements over 10 such intervals (20 h total) was summed for each individual cell. Cells were selected at hour zero only if they were clearly visible, and were discarded if, during the course of the measurement, they divided or died.

PLA quantification was performed by segmenting images by DAPI-positive nuclei. Each individual nucleus was considered individually, and the number of PLA punctae contained within each nucleus was measured in ImageJ through thresholding and the Analyze Particles function. Cells were separated for analysis not only by the antibodies used in the staining, but also by whether there was evidence of a transfection with the *Psme3*-FLAG construct. This was only discernible in cells in a condition containing the α-FLAG antibody, and so this distinction is not made in conditions that lacked this antibody.

The analysis of the number of myogenin-positive nuclei within each MHC-positive structure was performed manually in ImageJ (Schindelin et al, 2012). The total fraction of nuclei contained within MHC-positive structures was performed by thresholding the MHC channel to create a binary image, followed by gap filling and water shedding to cleanly separate the nuclei. DAPI-positive nuclei were similarly converted to a binary image, and the MHC image was used as a mask to subtract nuclei not contained within. The total number of nuclei remaining was quantified using the Analyze Particles function of ImageJ.

### Immunoblotting

Cells were collected by trypsinization and counted. 200 uL of RIPA buffer (50 mM Tris–HCl, pH 7.5, 150 mM NaCl, 1% Triton X-100, 0.5% sodium deoxycholate, 0.1% SDS, and 1x Pierce Protease Inhibitor Tablet [A32963; Thermo Fisher Scientific]) was added per $1 \times 10^6$ cells, which were incubated for 30 min at 4°C with rotation. Lysates were then briefly sonicated (three 5-s pulses at approximately 1 W) before being clarified by centrifugation for 8 min at 9,408$g$ at 4°C on a tabletop centrifuge. 4x sample loading buffer (1610747; Bio-Rad) with beta-mercaptoethanol was added to the lysates, which were then incubated at 95°C for 5 min.

Samples were loaded into 4–12% gradient gels (Bolt Bis-Tris Plus Mini Protein Gels, NW04122BOX; Thermo Fisher Scientific) and run in Bolt MOPS SDS (B000102; Thermo Fisher Scientific). Proteins were transferred using the Bio-Rad Trans-Blot Turbo Transfer System and were blocked in TBS-T plus 5% milk for 1 h before being incubated overnight with primary antibodies in the same solution. Membranes were washed three times in TBS-T before probing with secondary antibodies conjugated with HRP targeting either mouse (31430; Thermo Fisher Scientific) or rabbit (G-21234; Thermo Fisher Scientific). Membranes were incubated with SuperSignal West Pico

PLUS Chemiluminescent Substrate (34580; Thermo Fisher Scientific) before imaging.

### Immunoprecipitation

Cells were collected by trypsinization, washed once with PBS, and lysed for 30 min at 4°C with rotation in an immunoprecipitation (IP) buffer containing 20 mM Tris–HCl, pH 7.5, 100 mM NaCl, 1% NP-40, 1 mM EDTA, and 1x Pierce Protease Inhibitor Tablet (A32963; Thermo Fisher Scientific). 200 μl of IP buffer was used per $1 \times 10^6$ cells, and $2.5 \times 10^6$ cells were used for each immunoprecipitation. Cells were clarified by centrifugation for 8 min at 9,408$g$ at 4°C on a tabletop centrifuge. Primary antibodies were added directly into the clarified supernatant at 2.8 μg per $1 \times 10^6$ cells, with an isotype IgG (AB-105-C; Bio-Techne) used as a control. This solution was incubated overnight at 4°C with rotation. The next morning, 50 μl of Protein A Dynabeads (10008D; Thermo Fisher Scientific) was added directly to the solution, which was then incubated for 2 h at 4°C with rotation. Beads were removed from solution with a magnet and washed three times by resuspension with 200 μl of IP buffer. Beads were then collected in 100 μl of IP buffer and transferred to a clean tube before being resuspended in 3x sample loading buffer (1610747; Bio-Rad) with beta-mercaptoethanol and incubated for 5 min at 95°C. The supernatant was collected and frozen on dry ice before being stored at –80°C. The efficacy of the precipitation was assessed via immunoblot as described above, but using a protein A-HRP conjugate (18-160; Millipore) instead of the usual secondary antibodies to avoid detection of eluted IgG.

### CUT&RUN

CUT&RUN was performed using the kit provided by Cell Signaling Technology (#86652) according to the manufacturer's instructions, but with the following modifications. Three hundred thousand cells were used per condition. Furthermore, incubation was performed not overnight at 4°C as the manufacturer recommends, but rather for 30 min at room temperature to limit the level of cell death. Cells, unfixed, were frozen before assaying in 10% DMSO in FBS unless otherwise indicated.

Libraries were prepared using NEBNext Ultra II DNA Library Prep Kit for Illumina (E7645L) with primers from the NEBNext Multiplex Oligos for Illumina (Dual Index Primers Set 1, E7600S) according to the manufacturer's instructions. Prepared libraries were sent to Novogene for sequencing. Raw reads were cleaned with Trim Galore (http://www.bioinformatics.babraham.ac.uk/projects/trim_galore/) and aligned with Bowtie2, whereafter peaks were called with MACS2 (Zhang et al, 2008; Langmead & Salzberg, 2012). The threshold for determining the overlap of PSME3 and H3K4Me3 peaks was a single base pair. The permutation test used to determine the significance of their overlap was performed using the Bioconductor package RegioneR over 1,000 permutations.

### RNA sequencing

Cells were collected directly from the plate by washing once with PBS followed by the addition of TRIzol reagent (15596026; Thermo Fisher Scientific). The samples were incubated in the plate for 3 min

at room temperature before freezing in dry ice and long-term storage at −80°C. RNA was isolated from the sample using a chloroform extraction followed by purification with the RNeasy kit from QIAGEN. Samples were submitted to Novogene for library preparation and sequencing. RNA reads were aligned with RNA STAR, and comparative gene expression analysis was performed with DESeq2 (Dobin et al, 2013; Love et al, 2014).

### Mass spectrometry sample preparation

Samples in 3x Laemmli buffer were first cleaned up by SP3 using a commercial kit (PreOmics GmbH, 50 mg of beads per sample), then processed using the iST kit (PreOmics GmbH) according to the manufacturer's instructions. Tryptic digestion was stopped after 1 h, and cleaned-up samples were vacuum-dried. Finally, samples were redissolved by 10-min sonication in the iST kit's LC LOAD buffer.

### LC-MS/MS analysis

Samples were analyzed by LC-MS/MS on a nanoElute 2 nano-HPLC (Bruker Daltonics) coupled with a timsTOF HT (Bruker Daltonics), concentrated over a "Thermo Trap Cartridge 5 mm," and then bound to a PepSep XTREME column (1.5 $\mu$m C18-coated particles, 25 cm * 150 $\mu$m ID, Bruker P/N 1893476) heated at 50°C and eluted over the following 90-min gradient: solvent A, MS-grade $H_2O$ + 0.1% formic acid; solvent B, 100% acetonitrile + 0.1% formic acid; constant 0.60 nL/min flow; B percentage: 0 min, 2%; 90 min, 30%, followed immediately by a 8-min plateau at 95%. MS method: M/Z range = 99.993933-1700 Th, ion mobility range = 0.6–1.6 1/K0; transfer time = 60 $\mu$s, pre-pulse storage time = 12 $\mu$s, enable high sensitivity modus = off, ion polarity = Positive, scan mode = MS/MS (Pasef); TIMS parameters: ramp time = 100 ms, accumulation time = 100 ms; PASEF parameters: ms/ms scans = 10, total cycle time = 1.167166 s, charge range = 0-5, intensity threshold for scheduling = 2,000, scheduling target intensity = 15,000, exclusion release time = 0.4 min, reconsider precursor switch = on, current/previous intensity ratio = 4, exclusion window mass width = 0.015 m/z, exclusion window v·s/cm² width = 0.015 V*s/cm².

### LC-MS/MS data analysis

Raw files were searched in FragPipe version 20.0 against a *Mus musculus* proteome sourced from UniProtKB. Fixed cysteine modification was set to +57.02146 (Cysteine). Variable modifications were set to +15.9949 (methionine), +42.0106 (protein N-term), +79.96633 (STY), and −17.0265 (Gln -> pyroGlu). Peptide identifications were validated using Percolator. Results were filtered in Philosopher at the protein level at FDR 1%. MS1-level peptide quantitation was performed using IonQuant with match-between-runs turned on.

FragPipe's output was reprocessed using in-house R scripts, starting from the psm.tsv tables. MS1 intensities were renormalized to the median. The long format psm.tsv table was consolidated into a wide format peptidoform table, summing up quantitative values where necessary. Missing values were imputed using two different strategies: (i) the KNN (K-nearest neighbors) method for Missing-At-

Random values within sample groups, and (ii) the QRILC (Quantile Regression Imputation of Left-Censored data) method for Missing-Not-At-Random values. Peptidoform intensity values were renormalized as follows: 1 and 2. Peptidoform-level ratios were then calculated. Protein groups were inferred from observed peptides and quantified using an in-house algorithm, which (i) computes a mean protein-level profile across samples using individual, normalized peptidoform profiles ("relative quantitation" step), (ii) following the best-flyer hypothesis, normalizes this profile to the mean intensity level of the most intense peptidoform ("unscaled absolute quantitation" step); for protein groups with at least 3 unique peptidoforms, only unique ones were used; otherwise, razor peptidoforms were also included; phosphopeptidoforms and their unmodified counterparts were excluded from the calculations. Estimated expression values were log10-converted and renormalized using the Levenberg–Marquardt procedure. Average log10 expression values were tested for significance using a one-sided moderated *t* test per sample group (limma). Significance thresholds were calculated using the Benjamini–Hochberg procedure for false discovery rate values of 10%, 20%, and 30%. For all tests, regulated protein groups were defined as those with a significant *P*-value and a log2 ratio greater than 1 for immunoprecipitation experiments, and a log2 ratio greater than 0.3 for proteomics experiments. GO-term enrichment analysis was performed using Metascape Gene Annotation & Analysis Resource (Zhou et al, 2019).

## Data Availability

The CUT&RUN data from this publication have been deposited to the Gene Expression Omnibus (GEO) database and assigned the identifier GSE295256. The RNA-seq data from this publication have been deposited to the GEO database and assigned the identifier GSE294500. The co-immunoprecipitation mass spectrometry data have been deposited to the ProteomeXchange Consortium via the PRIDE (Perez-Riverol et al, 2025) partner repository with the dataset identifier PXD063339. The mass spectrometry proteomics data collected under PSME3 depletion have been deposited to the ProteomeXchange Consortium via the PRIDE partner repository with the dataset identifier PXD063351.

## Supplementary Information

## Acknowledgements

All proteomics analysis was done by the ISTA LSF Mass Spectrometry Service: Ewelina Dutkiewicz-Kopczynska processed the samples (digest and cleanup); Bella Bruszel optimized the acquisition methods, acquired the data, and performed all searches; and Armel Nicolas provided pre- and post-project consulting and post-processed the search results using a development version of their data analysis package, proteoCraft (publication pending). The authors would like to thank Saki for their clarity of thought and insight, as well as Dr. Lorenzo Puri and the members of his laboratory for

invaluable discussions relating to the project. This research was further supported by the Lab Support Facility and the Imaging and Optics Facility of ISTA.

## Author Contributions

KD Kuhn: conceptualization, formal analysis, investigation, visualization, methodology, and writing—original draft.

UH Cho: conceptualization, methodology, and writing—review and editing.

MW Hetzer: conceptualization, supervision, funding acquisition, and writing—review and editing.

## Conflict of Interest Statement

The authors declare that they have no conflict of interest.

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
