## [Reviewer comments · Life Science Alliance]

Life Science Alliance

PSME3 Regulates Migration and Differentiation of Myoblasts

Kenneth D Kuhn, Ukrae H Cho and Martin Hetzer

DOI: <https://doi.org/10.26508/lsa.202503208>

Corresponding author(s): *Dr. Martin Hetzer (Institute of Science and Technology Austria)*

Review Timeline:

Submission Date:	2025-01-07
Editorial Decision:	2025-02-03
Revision Received:	2025-04-27
Editorial Decision:	2025-05-29
Revision Received:	2025-06-03
Accepted:	2025-06-04

Scientific Editor: Tim Fessenden

Transaction Report:

February 3, 2025

Re: Life Science Alliance manuscript #LSA-2025-03208

Dr. Martin Hetzer
Institute of Science and Technology Austria
Am Campus 1
Klosterneuburg, NO 3400
Austria

Dear Dr. Hetzer,

Thank you for submitting your manuscript entitled "PSME3 Regulates Migration and Differentiation of Myoblasts" to Life Science Alliance. The manuscript was assessed by expert reviewers, whose comments are appended to this letter. We invite you to submit a revised manuscript addressing the Reviewer comments.

Thank you for this interesting contribution to Life Science Alliance. We are looking forward to receiving your revised manuscript.

Sincerely,

B. MANUSCRIPT ORGANIZATION AND FORMATTING:

Reviewer #1 (Comments to the Authors (Required)):

In this manuscript, the authors investigated the role of the on-canonical proteasome activator PSME3 during the skeletal myogenesis, using the model of C2C12 cultures of murine myoblasts. The authors first show that PSME3 binds to promoters, but do not globally affect mRNA transcription from bound promoters. Then, they show that ectopically expressed PSME3 interacts with the HSP90 co-chaperone NUDC in the nucleus. In a separate assay they found that cells lacking PSME3 exhibit an increased migration rate and higher abundance of several cell adhesion proteins implicated in cytoskeletal organization, migration and adhesion. Finally, it is shown that PSME3 is important for muscle fusion, likely in facilitates myoblast differentiation in a proteasome-independent manner.

This study could be interesting as it might assign a role to PMSE3 in the regulation of skeletal myogenesis; however, as they stand the data are mostly descriptive and the major findings - i.e. PMSE3 binding to gene promoters; PMSE3 interaction with HSP90 in the nucleus, regulation of cell motility/adhesion; PMSE3 regulation of myoblast fusion - remain unconnected, leaving this reviewer with no final conclusion on the mechanism by which nuclear PMSE3 regulates skeletal myogenesis.

Specific Points:

1. The authors should determine the functional link that connects PMSE3 binding to gene promoters, PMSE3 interaction with HSP90 in the nucleus, regulation of cell motility/adhesion and myoblast fusion.
2. PMSEE binding to gene promoter does not seem to alter gene expression. This evidence leaves this reviewer with the question of what is the function of promoter-bound PMSE3?
3. Nuclear interactions between PSME3 and RPRD1A or NUDC are detected only after over-expression of FLAG-PMSE3, which might lead to artifactual results. Can the authors try to detect interactions between endogenous proteins?
4. Despite the detection of interactions with nuclear partners that predict transcriptional and/or post-translational mechanism of regulation of gene/proteins implicated in cell motility, adhesion and fusion (the biological processes affected by PMSE3 knockdown) it remains unclear the mechanism and level of regulation of these processes by PMSE3.

Reviewer #2 (Comments to the Authors (Required)):

The manuscript provides novel and significant insights into the role of PSME3 in myoblast differentiation, cell migration, and adhesion. The findings, particularly the proteasome-independent function of PSME3 and its interaction with NUDC, are valuable contributions to the field. However, several areas require additional experimental support or clarification to enhance the depth and impact of the work. Below are my detailed comments and suggestions:

Major Points

The observation that PSME3 binds to active promoters during cycling and loses this binding during differentiation is intriguing but lacks an explanation of its biological significance. One approach to address this would be to perform CUT&RUN or ChIP-seq at additional differentiation time points (e.g., Day 0, Day 1, Day 2, and Day 3) to clarify the temporal dynamics of PSME3's binding. This could provide valuable insights into how PSME3 might regulate chromatin states during differentiation.

The interaction between PSME3 and NUDC is well-demonstrated, but its functional significance in regulating cell adhesion and migration needs further clarification. To strengthen this aspect, the authors could consider performing siRNA or CRISPR-mediated knockdown of NUDC in PSME3-depleted cells and evaluating changes in cell migration and adhesion protein levels (e.g., MYADM and ITGB3). This would help establish whether NUDC mediates the observed effects of PSME3 on cell migration and adhesion.

Minor Points

It might be helpful to include additional details on the statistical methods used in key figures (e.g., Figures 3C, 4E), such as the type of test performed and the criteria for determining significance. This could improve the clarity and rigor of the presented results.

Quantifying the proximity ligation assay (PLA) signals shown in Figures 2 and 3 could provide a more robust demonstration of PSME3's interactions with RPRD1A and NUDC. Including information on how these interactions change under different conditions or at various stages of differentiation could also enhance the impact of the findings.

Expanding the discussion of the functional implications of increased adhesion protein levels, particularly MYADM and ITGB3, in PSME3-depleted cells could strengthen the manuscript. Speculating on how these changes might impact cell migration and differentiation would provide additional context for the observed results.

The evidence supporting PSME3's preferential binding to highly active promoters in Figure 1 could be clarified further. Including detailed annotations or statistical comparisons with appropriate controls might help contextualize the findings for the readers.

Thank you for the opportunity to submit a revised version of our manuscript, and for transmitting the reviewer's thoughtful feedback. We appreciate their efforts in helping us strengthen our study. Please find below a point-by-point response to the comments they provided.

Reviewer #1

"1. The authors should determine the functional link that connects PMSE3 binding to gene promoters, PMSE3 interaction with HSP90 in the nucleus, regulation of cell motility/adhesion and myoblast fusion."

We agree that uncovering such a link would greatly strengthen our study, and have sought to do so while carrying out our research. While we have performed additional experiments to establish a connection between PSME3's association with NUDC and its regulation of migration and myogenesis (shown in the new Figure 5 and described in further detail below), we believe that the DNA-binding activity of PSME3 may hold little functional relevance for these activities. Instead, DNA-bound PSME3 may be engaged in some additional function which our experiments lack the power to resolve, or is not active during the myogenic process. We have, however, performed additional experiments and analyses, detailed in the remainder of this letter, in which we attempt to provide a more integrated view of PSME3's activity than the original manuscript afforded.

"2. PMSE3 binding to gene promoter does not seem to alter gene expression. This evidence leaves this reviewer with the question of what is the function of promoter-bound PMSE3?"

In the discussion section of our manuscript, we suggest that PSME3 may be primed to perform repair in response to DNA damage. However, we have not undertaken experiments to verify this hypothesis as it lacks relevance for the differentiation process, which is the focus of our study. Instead, we sought to determine whether DNA-bound PSME3 influences myogenesis through a mechanism independent of gene expression regulation.

It has previously been shown that PSME3 may facilitate the recruitment of splicing machinery to transcribed genes (Baldin et al., 2008). Changes in splicing might conceivably affect myogenesis, so we employed an alternative splicing analysis across differentiation (Rebuttal Figures 1 through 3). Overall, we found that loss of PSME3 had no global effect on any individual splicing process (Panel A of Rebuttal Figures 1 through 3). However, at each timepoint we found a handful of differentially spliced genes (Panel B of Rebuttal Figures 1 through 3). Among these was Rac1b, a highly active variant of the cytoskeletal regulator Rac1 (Matos et al., 2003; Melzer et al., 2019), which sharply decreased in utilization in PSME3-depleted Day 2 cells (Rebuttal Fig. 4 A and B). Properly-tuned Rac1 activity is critical for myogenesis (Melzer et al., 2019; Vasyutina et al., 2009), and we reasoned that alteration in this transcript may explain the differentiation phenotype we observed. We found that depletion of Rac1 by siRNA (Rebuttal Fig. 4 C) was sufficient to produce a differentiation defect qualitatively

similar to loss of PSME3 (Rebuttal Fig. 4 D). However, Rac1b levels did not change significantly over the course of differentiation following PSME3 depletion as detected by Western blot (Rebuttal Fig. 4 E) or by mass spectrometry (see source data). While we cannot rule out the possibility that PSME3-mediated splicing of a few select transcripts may mediate PSME3's effect on cellular differentiation, PSME3's major function at the DNA in our system does not appear to be in the regulation of gene splicing. As a result, what the purpose of DNA-bound PSME3 is, we cannot firmly state.

“3. Nuclear interactions between PSME3 and RPRD1A or NUDC are detected only after over-expression of FLAG-PMSE3, which might lead to artifactual results. Can the authors try to detect interactions between endogenous proteins?”

We would first point to Figure 2 A-B of the manuscript where interactions between the endogenous PSME3 population with NUDC and RPRD1A were detected. However, one might argue that spurious interactions can occur in an immunoprecipitation lysate, and so identification of antibodies suitable for detection of endogenous PSME3 for PLA would be desirable. Unfortunately, in our hands, the only antibodies suitable for detection of endogenous PSME3, NUDC, and RPRD1A by immunofluorescence are all derived from rabbit, and so cannot be combined for a PLA experiment. However, to provide more information about the localization of PSME3, we have included an additional panel in our manuscript (Fig. S3 A) that shows staining of the endogenous PSME3 pool, which can be seen to be nuclear and which largely resembles staining of FLAG-PSME3 (Fig. 3 C).

“4. Despite the detection of interactions with nuclear partners that predict transcriptional and/or post-translational mechanism of regulation of gene/proteins implicated in cell motility, adhesion and fusion (the biological processes affected by PMSE3 knockdown) it remains unclear the mechanism and level of regulation of these processes by PMSE3.”

While our original manuscript showed only mechanisms PSME3 does not employ to regulate differentiation and migration (i.e., not through the proteasome, not through gene expression), the addition of PSME3/NUDC double knockdown experiments provides a potential mechanism for PSME3's activity. These results show that PSME3 is dependent on NUDC for its ability to regulate migration and myogenesis, and may therefore act upstream of NUDC and influence its function. These findings are summarized in Figure 5 of the revised manuscript.

Reviewer #2

“The observation that PSME3 binds to active promoters during cycling and loses this binding during differentiation is intriguing but lacks an explanation of its biological significance. One approach to address this would be to perform CUT&RUN or ChIP-seq at additional differentiation time points (e.g., Day 0, Day 1, Day 2, and Day 3) to clarify the temporal dynamics of PSME3's binding. This could provide valuable insights into how PSME3 might regulate chromatin states during differentiation.”

We agree that it would be interesting to understand whether there exists a sequential or hierarchical pattern of PSME3 binding over time, such that some regions might lose PSME3 more rapidly than others over differentiation. Because PSME3's DNA-binding activity does not seem to be relevant for its regulation of myogenesis, we believe this to be beyond the scope of our study. However, we have made several attempts to understand which genes PSME3 preferentially associates with, and to determine whether this binding has any connection to the differentiation process. One avenue we pursued was to determine whether PSME3 is bound to promoters which undergo extensive changes in activity during differentiation. While we found that PSME3-bound genes were found to be more resistant to changes in expression over differentiation (Rebuttal Figure 5 A-C), PSME3-binding explains less than 1% of total variance in Log2FCR between Cycling and Day 2 cells. Therefore PSME3's DNA-binding activity seems to be of limited relevance for myogenesis.

“The interaction between PSME3 and NUDC is well-demonstrated, but its functional significance in regulating cell adhesion and migration needs further clarification. To strengthen this aspect, the authors could consider performing siRNA or CRISPR-mediated knockdown of NUDC in PSME3-depleted cells and evaluating changes in cell migration and adhesion protein levels (e.g., MYADM and ITGB3). This would help establish whether NUDC mediates the observed effects of PSME3 on cell migration and adhesion.”

As requested, we performed a Western blot on cells deficient in either PSME3, NUDC, or both concurrently and stained for ITGB3, a major cell adhesion-regulating protein identified in the manuscript, as well as ITGAV, one of its major binding partners, and a protein which showed significant through sub-threshold changes by mass spectrometry (a suitable MYADM antibody could not be found). Unfortunately, the blot was inconclusive and no clear changes in protein level were discernible. However, we made use of these cells to perform several functional assays, which now compose Figure 5 of the revised manuscript. We demonstrate that, in the regulation of both migration and differentiation, PSME3 likely acts upstream of NUDC. Loss of NUDC results in a decrease in migration and differentiation efficiency, and additional loss of PSME3 is neither able to increase migration rates or further impair myogenesis. This indicates that PSME3 may act as a regulator of NUDC activity, perhaps impairing its ability to stabilize migration- and adhesion-related proteins, and that PSME3 becomes dispensable for these processes in the absence of NUDC.

“It might be helpful to include additional details on the statistical methods used in key figures (e.g., Figures 3C, 4E), such as the type of test performed and the criteria for determining significance. This could improve the clarity and rigor of the presented results.”

Figure legends were adjusted to explicitly state the statistical test performed, the number of replicates, and the *n*. Where necessary, additional explanation was added to the Methods section.

“Quantifying the proximity ligation assay (PLA) signals shown in Figures 2 and 3 could provide a more robust demonstration of PSME3's interactions with RPRD1A and NUDC. Including

information on how these interactions change under different conditions or at various stages of differentiation could also enhance the impact of the findings.”

Quantification of the PLA signals for the interaction of PSME3 with both NUDC and RPRD1A were added to their respective figures of the manuscript (now Fig. S2 and Fig. S4)

“Expanding the discussion of the functional implications of increased adhesion protein levels, particularly MYADM and ITGB3, in PSME3-depleted cells could strengthen the manuscript. Speculating on how these changes might impact cell migration and differentiation would provide additional context for the observed results.”

An expanded discussion of the potential role of ITGB3 and MYADM in migration and differentiation were added to the discussion section (the second to last major paragraph).

“The evidence supporting PSME3’s preferential binding to highly active promoters in Figure 1 could be clarified further. Including detailed annotations or statistical comparisons with appropriate controls might help contextualize the findings for the readers.”

A permutation test to determine the statistical significance of the overlap between PSME3 and H3K4Me3 peaks was performed; the results were added to Fig. 1 B.

Finally, the raw data from the RNA sequencing, CUT&RUN, and both mass spectrometry experiments have been submitted to public databases, as described in the Data Availability section of the revised manuscript. The mass spectrometry experiments were recently submitted and will be made publicly accessible immediately. However, should the reviewers need to access this data before this change takes place, they can use the following login information on the PRIDE database (<https://www.ebi.ac.uk/pride/>):

PSME3 co-immunoprecipitation data:

Project accession: PXD063339

Token: GM9bUKJFFZdf

Proteomics under PSME3-depletion data:

Project accession: PXD063351

Token: AKwR2WfMYvNN

We thank the reviewers for their comments and hope you will find these changes sufficient for approval of the manuscript’s publication. We remain available for any further correspondence concerning our study.

Sincerely,

Martin Hetzer

A

B

Rebuttal Figure 1: Loss of PSME3 does not result in global splicing changes in cycling C2C12 cells. **A)** RNA-sequencing data was subjected to analysis by IsoformSwitchAnalyzerR to measure global changes in splicing patterns. **B)** A volcano plot showing the change in isoform

utilization between control and PSME3 knockdown cells (abbreviated dIF, which is the difference in the abundance of a splicing variant as a fraction of the total number transcripts) and the significance of the change. Several altered transcripts are labeled with gene names; not all differentially spliced isoforms are expected to show functional differences.

A

B

Rebuttal Figure 2: Loss of PSME3 does not result in global splicing changes in Day 0 C2C12 cells. A) RNA-sequencing data was subjected to analysis by IsoformSwitchAnalyzerR to

measure global changes in splicing patterns. **B)** A volcano plot showing the change in isoform utilization between control and PSME3 knockdown cells (abbreviated dIF, which is the difference in the abundance of a splicing variant as a fraction of the total number transcripts) and the significance of the change. Several altered transcripts are labeled with gene names; not all differentially spliced isoforms are expected to show functional differences.

A

C2C12 Cells Day 2 of Differentiation

C2C12 Cells Day 2 of Differentiation

B

Rebuttal Figure 3: Loss of PSME3 does not result in global splicing changes in Day 2 C2C12 cells. **A)** RNA-sequencing data was subjected to analysis by IsoformSwitchAnalyzeR to measure global changes in splicing patterns. **B)** A volcano plot showing the change in isoform utilization between control and PSME3 knockdown cells (abbreviated dIF, which is the difference in the abundance of a splicing variant as a fraction of the total number transcripts) and the significance of the change. Several altered transcripts are labeled with gene names; not all differentially spliced isoforms are expected to show functional differences.

Rebuttal Figure 4: Rac1b is critical for differentiation, but is not affected at the protein level by loss of PSME3. **A)** A splicing diagram showing the addition of exon #4 in the Rac1b transcript that is not found in Rac1. **B)** An analysis of isoform usage performed with IsoformSwitchAnalyzerR showing a reduced usage of Rac1b by PSME3-deficient Day 2 C2C12 cells. **C)** Western blot of Day 0 C2C12 cells treated with one of several siRNA molecules targeting Rac1b, PSME3, or a scrambled sequence. **D)** Day 2 C2C12 cells depleted of Rac1b by siRNA molecule #2 were fixed and stained for MHC (green) and myogenin (purple). Scale bar is 200 microns in length. **E)** Western blot of C2C12 cells across differentiation treated with either an siRNA molecule targeting PSME3 or a scrambled sequence.

Rebuttal Figure 5: PSME3-bound genes are slightly more resistant to expression changes over differentiation. **A)** A volcano plot showing the magnitude of change of individual transcripts across differentiation and color-coded according to whether the gene is bound by PSME3 in cycling C2C12 cells. The x-axis showing the \log_2 fold-change ratio in expression at Day 2 of differentiation compared to Cycling cells. **B)** A violin plot showing the distribution of the fold-change ratio of individual transcript at Day 2 versus Cycling cells, grouped by whether that gene is bound by PSME3 in Cycling cells. P value was generated using Welch's two sample t-test. Note that data from both panels is drawn from siScr-treated C2C12 cells represented in Figure 2 D of the manuscript. **C)** A violin plot showing the magnitude (absolute value) of the fold-change ratio of individual transcripts at Day 2 versus Cycling cells, divided by whether a gene is bound by PSME3 in cycling cells. P value was generated using Welch's two sample t-test.

Bibliography

- Baldin V, Militello M, Thomas Y, Doucet C, Fic W, Boireau S, Jariel-Encontre I, Piechaczyk M, Bertrand E, Tazi J, Coux O. 2008. A novel role for PA28gamma-proteasome in nuclear speckle organization and SR protein trafficking. *Mol Biol Cell* 19:1706–1716. doi:10.1091/mbc.e07-07-0637
- Matos P, Collard JG, Jordan P. 2003. Tumor-related alternatively spliced Rac1b is not regulated by Rho-GDP dissociation inhibitors and exhibits selective downstream signaling. *J Biol Chem* 278:50442–50448. doi:10.1074/jbc.M308215200
- Melzer C, Hass R, Lehnert H, Ungefroren H. 2019. RAC1B: A Rho GTPase with Versatile Functions in Malignant Transformation and Tumor Progression. *Cells* 8. doi:10.3390/cells8010021
- Vasyutina E, Martarelli B, Brakebusch C, Wende H, Birchmeier C. 2009. The small G-proteins Rac1 and Cdc42 are essential for myoblast fusion in the mouse. *Proc Natl Acad Sci USA* 106:8935–8940. doi:10.1073/pnas.0902501106

May 29, 2025

RE: Life Science Alliance Manuscript #LSA-2025-03208R

Dr. Martin Hetzer
Institute of Science and Technology Austria
Am Campus 1
Klosterneuburg, NO 3400
Austria

Dear Dr. Hetzer,

Thank you for submitting your revised manuscript entitled "PSME3 Regulates Migration and Differentiation of Myoblasts". As you will see, reviewers do make any further requests and are overall satisfied with your revised manuscript. We would be happy to publish your paper in Life Science Alliance pending final revisions necessary to meet our formatting guidelines.

- Please add the X and Bluesky handles of your host institute/organization, as well as your own and/or one of the authors, in our system.
- Please rename "Bibliography" to References.
- Please use the [10 author names, et al.] format in your references (i.e., limit the author names to the first 10).
- Please add your main and supplementary figure legends to the main manuscript text after the references section.
- It is recommended to exclude figures from the manuscript text and upload them separately.
- Please add molecular weight markers to the blots in Figure 4B.

A. FINAL FILES:

B. MANUSCRIPT ORGANIZATION AND FORMATTING:

Sincerely,

Reviewer #1 (Comments to the Authors (Required)):

While the authors did not completely address the points raised by reviewers at the experimental level, this referee understands the provocative and speculative nature of the findings reported in this manuscript and agrees that further investigation might be needed to address mechanistic insights behind the proposed role of PSME as a regulator of gene expression. I therefore leave to the editor the final decision to accept the manuscript as it is.

Reviewer #2 (Comments to the Authors (Required)):

I have carefully reviewed the revised manuscript and the authors' responses to my comments. I am satisfied that the authors have adequately addressed all of my concerns and have substantially improved the quality and clarity of the paper. Therefore, I recommend this manuscript for acceptance.

June 4, 2025

RE: Life Science Alliance Manuscript #LSA-2025-03208RR

Dr. Martin Hetzer
Institute of Science and Technology Austria
Am Campus 1
Klosterneuburg, NO 3400
Austria

Dear Dr. Hetzer,

Thank you for submitting your Research Article entitled "PSME3 Regulates Migration and Differentiation of Myoblasts". It is a pleasure to let you know that your manuscript is now accepted for publication in Life Science Alliance. Congratulations on this interesting work.

DISTRIBUTION OF MATERIALS:

Again, congratulations on a very nice paper. I hope you found the review process to be constructive and are pleased with how the manuscript was handled editorially. We look forward to future exciting submissions from your lab.

Sincerely,
